# FicGCN: Unveiling the Homomorphic Encryption Efficiency from Irregular Graph Convolutional Networks

**Zhaoxuan Kan** [1 2 3] **Husheng Han** [1 2] **Shangyi Shi** [1 2] **Tenghui Hua** [1 2] **Hang Lu** [1 3] **Xiaowei Li** [1 3] **Jianan Mu** [1]
**Xing Hu** [1 3 4]

## Abstract

Graph Convolutional Neural Networks (GCNs) have gained widespread popularity in various fields like personal healthcare and financial systems, due to their remarkable performance. Despite the growing demand for cloud-based GCN services, privacy concerns over sensitive graph data remain significant. Homomorphic Encryption (HE) facilitates Privacy-Preserving Machine Learning (PPML) by allowing computations to be performed on encrypted data. However, HE introduces substantial computational overhead, particularly for GCN operations that require rotations and multiplications in matrix products. The sparsity of GCNs offers significant performance potential, but their irregularity introduces additional operations that reduce practical gains. In this paper, we propose FicGCN, a HE-based framework specifically designed to harness the sparse characteristics of GCNs and strike a globally optimal balance between aggregation and combination operations. FicGCN employs a latency-aware packing scheme, a Sparse Intra-Ciphertext Aggregation (SpIntra-CA) method to minimize rotation overhead, and a region-based data reordering driven by local adjacency structure. We evaluated FicGCN on several popular datasets, and the results show that FicGCN achieved the best performance across all tested datasets, with up to a $4.10\times$ improvement over the latest design.

[1]State Key Lab of Processors, Institute of Computing Technology, Chinese Academy of Sciences, Beijing, China [2]University of Chinese Academy of Sciences, Beijing, China [3]Zhongguancun Laboratory, Beijing, China [4]Shanghai Innovation Center for Processor Technologies, Shanghai, China. Correspondence to: Jianan Mu <mujianan@ict.ac.cn>.

*Proceedings of the 42nd International Conference on Machine Learning*, Vancouver, Canada. PMLR 267, 2025. Copyright 2025 by the author(s).

## 1. Introduction

Graph-based machine learning has gained significant prominence across numerous domains, with Graph Convolutional Neural Networks (GCNs) (Kipf & Welling, 2016) emerging as a key paradigm demonstrating superior performance in various applications, including human action recognition (Si et al., 2018; Yan et al., 2018), financial recommendation systems (Wu et al., 2022), and drug discovery (Bongini et al., 2021). As the demand for cloud-based GCN services continues to grow, privacy concerns surrounding sensitive graph data have become especially critical, given that graph data typically include large amounts of confidential information.

Privacy-Preserving Machine Learning (PPML) using Homomorphic Encryption (HE) offers a promising solution to alleviate these concerns by allowing clients to encrypt their data before sending it to the cloud server. This setup ensures that the server can perform computation directly on ciphertexts without ever decrypting them, effectively safeguarding user privacy. Specifically, in a privacy-preserving GCN cloud service, the server holds the well-trained weight matrices $W$ and adjacency matrix $A$ which are all plaintexts. The client sends the encrypted data to the server (Ran et al., 2022) thus preventing data leakage to the server. Although this framework successfully protects sensitive graph embeddings, it also introduces substantial overhead.

When a Cheon-Kim-Kim-Song (CKKS) (Cheon et al., 2017) scheme is employed, the computational overhead increases a lot, often resulting in performance degradation by a factor of over three orders. Two primary factors contribute to this overhead. First, due to the requirements for security and correctness, homomorphic ciphertexts are much larger than the original data size, and the operation complexity is increased. Second, for batched CKKS ciphertext, moving and accumulating data within a vector, require expensive homomorphic ciphertext rotation operation. This results in high overhead for matrix multiplication on the ciphertext.

For HE-GCN computation, the bottleneck in computation speed still lies in performing the aggregation and combination operations, specifically the plaintext-ciphertext matrix multiplication $AXW$ where $A$ and $W$ are plaintexts and $X$

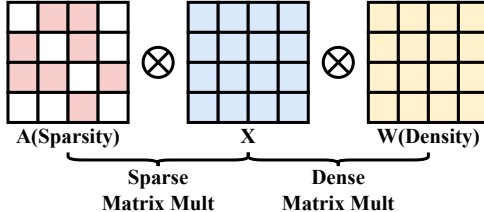

(a) Inconsistency in computation patterns.

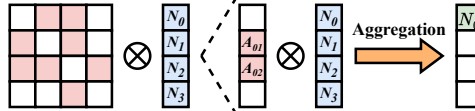

(b) Irregularity in neighbor aggregation patterns.

*Figure 1.* The contradiction between the computing characteristics of GCN and CKKS.

is ciphertext, which includes a large number of rotations and multiplications. Drawing inspiration from the efficiency optimization of plaintext GCN computations, recent work leverages the sparsity of $A$ for optimization. To this end, an adjacency-matrix-aware data packing and multiplication method is proposed (Ran et al., 2022), which reduces the rotation overhead in the $X' = AX$ process by exploiting the irregular sparsity of $A$. Unluckily, due to the irregular sparsity of $A$, additional SIMD multiplication operations are introduced when calculating $X' = AX$, which results in limited overall speedup. Therefore, the main bottleneck in improving the computational efficiency of existing designs lies in how to leverage the irregular sparsity of GCN computations while adhering to the SIMD calculation pattern of HE, ultimately reducing the computational overhead.

Addressing this challenge is difficult, as GCN's irregular operational patterns stand in stark contrast to HE's SIMD-based data processing. We analyze the conflicts as follows: **1. Inconsistency in computation patterns.** As shown in Figure 1(a), in $AXW$, the left multiplication of $X$ by $A$ is calculating sparse weighted aggregation between a partial set rows of $X$. The right multiplication of $X$ by $W$ is dense, achieving weighted aggregation between all columns of $X$. This introduces the conflict: for dense multiplication, fully-packed ciphertexts can improve efficiency, while for sparse multiplication, fully-packed ciphertexts lead to redundant rotations and multiplications. Additionally, The best data packing of $X$ required for left multiplication and right multiplication is also different. **2. Irregularity in neighbor aggregation patterns.** As shown in Figure 1(b), in the calculation of $AX$, since the adjacency matrix $A$ is sparse and irregular, each node in the graph requires performing different aggregation patterns. In CKKS ciphertexts, these nodes are encrypted into one vector. Therefore, this necessitates rotating the corresponding nodes and summing them.

However, the contradiction between irregular sparsity and SIMD makes it challenging to balance ciphertext utilization and minimize the number of rotations. When using packed ciphertexts, rotation is required for each node's specific aggregation needs, leading to significant computational overhead.

In this paper, we propose FicGCN, a CKKS-based framework designed to exploit the irregular sparsity of GCNs. To resolve the mismatch between the aggregation and combination operations, we devise a latency-aware packing scheme that strikes a globally optimal balance for overall performance. For the irregular sparsity of $A$ within aggregation, we employ a two-pronged strategy: first, a Sparse Intra-Ciphertext Aggregation (SpIntra-CA) technique that leverages HE's operational properties to minimize total rotation overhead; second, a region-based data reordering based on the structural local adjacency of $A$. We implement and evaluate these designs, and our experimental results show that FicGCN achieves up to $4.10\times$ improvement on different datasets compared to the latest work. Our contributions can be summarized as follows:

- We propose an optimal layer-wise aggregation scheduling strategy based on the data dimensions, model structure, and the latency of different HE operations. This enables efficient inference for data of various scales.

- We propose a sparse intra-ciphertext rotation technique and a region-based data reordering to minimize total rotation overhead in aggregation.

- We evaluate FicGCN on four popular datasets and the results show that FicGCN achieved the best performance across all tested datasets, with up to a $4.10\times$ improvement over the latest design.

## 2. Preliminary

### 2.1. CKKS Homomorphic Encryption Scheme

CKKS (Cheon et al., 2017), as a popular HE scheme, has been widely employed in confidential neural network inference since its ability for floating-point numbers encryption and computing using a scaling factor $\Delta$. In CKKS, a ciphertext $\mathbf{c} \in \mathcal{R}_Q^2$ can be decrypted by computing $\mathbf{c} \cdot sk \bmod Q = \mathbf{m} + \mathbf{e}$, where $\mathcal{R}_Q = \mathbb{Z}_Q[X]/X^N + 1$ is the residue cyclotomic polynomial ring and $sk$ refers to secret keys held by the client, e is a small error that provides security. The modulus is $Q = \prod_{i=1}^{L} q_i$, where $L$ is the multiplication level of the ciphertext. Each time a homomorphic multiplication is performed, $L$ decreases by one, making the next homomorphic operation faster. A ciphertext has N/2 slots to accommodate N/2 complex numbers and it supports homomorphic addition, multiplication and rotation:

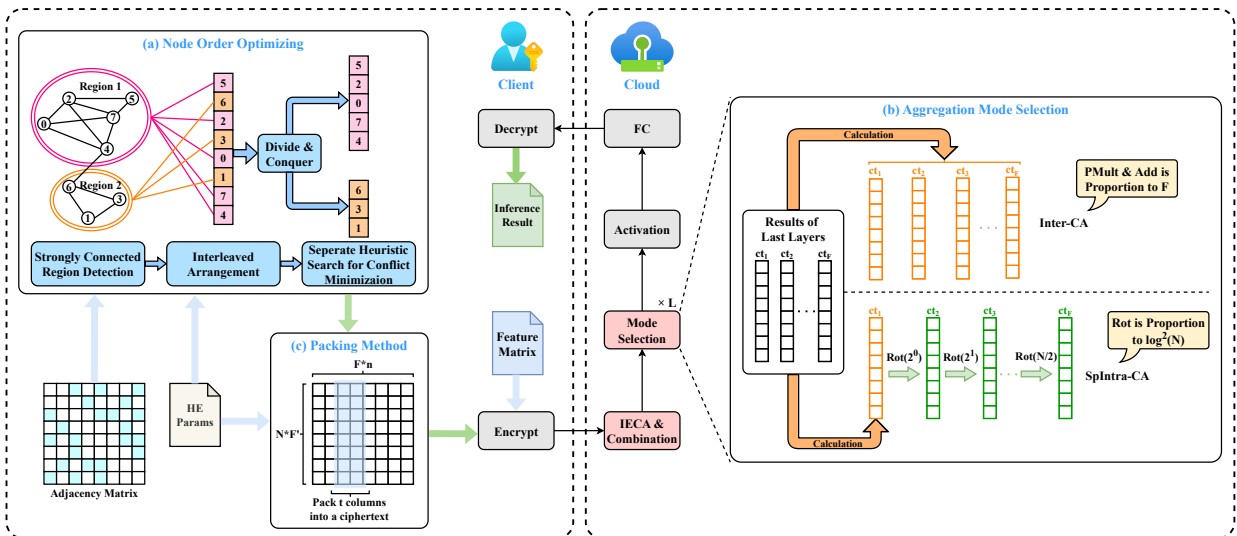

*Figure 2.* Workflow of FicGCN. (a) Three main steps of the HE-specific reordering algorithm: Detect, Arrange and Search. (b) The computational complexity comparison between Inter-CA and SpIntra-CA, SpIntra-CA encompasses $log(N)$ fundamental rotations along with several additional rotations triggered by conflicts. *(The dashed lines represent optional data paths, and the modules shown in detail are those that represent the optimization achieved through our innovative HE graph computation scheduling strategy, while the white modules retain the SOTA structure.)*

$Dec(\mathbf{c}_1 \oplus \mathbf{c}_2) = \mathbf{m}_1 \oplus \mathbf{m}_2$ ; $Dec(\mathbf{c}_1 \otimes \mathbf{c}_2) = \mathbf{m}_1 \otimes \mathbf{m}_2$; $Dec(\mathbf{m} \otimes \mathbf{c}_1) = \mathbf{m} \otimes \mathbf{m}_1$ ; $Rot(Enc(v_0, ..., v_{\frac{N}{2}-1}), k) = Enc(v_k, ..., v_{\frac{N}{2}-1}, v_0, ..., v_{k-1})$

### 2.2. Graph Convolution Network and GraphSage

**GCN** contains multiple linear layers (consecutive matrix multiplications, GCNConv) and nonlinear layers. In GCN-Conv, there are two stages, Aggregation and Combination, corresponding to the left and right multiplications involving the feature matrix $X$. The layer-wise forward propagation can be expressed as follows (Kipf & Welling, 2016):

$$X^{l+1} = GCNConv(X^l) = \sigma(\hat{A}X^lW^l)$$

where $\hat{A}$ is normalized adjacency matrix , $W^l$ is the weight matrix for feature dimension transformation in $l^{th}$ layer, $\sigma(\cdot)$ denotes the non-linear activation function.

**GraphSage** (Hamilton et al., 2017) is a popular GCN scheme that introduces neighbor sampling in Aggregation, allowing each node's update to depend only on features from a subset of neighboring nodes. This significantly reduces the computational complexity while maintaining inference accuracy. The forward propagation chosen in this paper is:

$$X_v^{l+1} = \sigma(MEAN(\{X_v^l\} \cup \{X_u^l, u \in \mathcal{N}eighbor(v)\}) \cdot W^l)$$

### 2.3. Related Work

GCN inference under HE is an effective method commonly used in cloud computing scenarios to protect privacy-sensitive data, such as clients' fingerprint information, medical data, and financial transaction records (Choi et al., 2024; Yan et al., 2018; Matsunaga et al., 2019). Early cloud computing was mainly integrated with CNN. CryptoNets (Gilad-Bachrach et al., 2016) is the first to introduce HE into cloud computation. Due to the time overhead gap of $10^6\times$ between plaintext and ciphertext, existing work such as HEMET (Lou & Jiang, 2021) made a trade-off between model parameters and encryption parameters to utilize mobile networks. (Lee et al., 2022) also observed the parallelism between CNN channels and leveraged this to effectively combine with the SIMD property of CKKS but still cost 6351 seconds when predicting on 50 images in CIFAR-100.

The most distinguishing feature of GCN compared to other types of networks is its inherent sparsity. The plaintext inference has direct access to the feature vectors of the nodes, which enables fully exploiting their sparsity based on adjacency information, thereby simplifying the computation. (Jia et al., 2020)also identified common aggregation patterns and reused them, further enhancing cache performance and reducing computational overhead. Multi-Party secure Computation (MPC) methods (Reagen et al., 2021; Hao et al., 2022; Zeng et al., 2023a;b; Wu et al., 2024) are also introduced in confidential GCN to simulate the property of arbitrary node access in plaintext (Srinivasan et al., 2019; Reagen et al., 2021; Hao et al., 2022), but face significant communication overhead and rely on client computation.

Recent works tend to optimize the performance but fail

to fully utilize the sparsity. Gazelle (Juvekar et al., 2018) was the first to apply diagonal encoding in the optimization of HE matrix multiplication, enhancing computational efficiency. Peinguin (Ran et al., 2024) adopted this idea by leveraging block-wise packing of $X$ at the first layer, thereby achieving a trade-off when computing $AX$ and $XW$. However, it had no utilization of the graph sparsity. CryptoGCN (Ran et al., 2022) made full use of the sparsity of $A$ but it achieved suboptimal efficiency for requiring redundant computations to obtain the multiple ciphertexts for aggregation in the next layer.

Besides, recent works like LinGCN, THE-X and HETAL (Lee et al., 2023; Peng et al., 2024; Ao & Boddeti, 2024; Jha et al., 2021; Ran et al., 2023; Chen et al., 2022) have observed that the nonlinear layers or feature maps are not equally important, and have proposed algorithms to prune them. Meanwhile, prior works like FedML-HE (Jin et al., 2023) and BatchCrypt (Zhang et al., 2020) enable clients to strategically compromise certain node features' security via Selective Encryption and reduce inference latency via parameter quantization. Corresponding technologies above are orthogonal and can be compatible with our work.

### 2.4. Threat Model

The threat model of this paper is similar to CryptoGCN. The server holds the well-trained weight matrices $W$ and adjacency matrix $A$ which are all plaintexts. The client sends the encrypted data to the honest but curious server and holds the secret keys thus preventing data leakage to the server. After calculating on encrypted data, the server sends the encrypted outcomes back to the client. The client can then decrypt them and get the results.

### 2.5. Security Analysis

The homomorphic encryption scheme employed in this study—CKKS derives its security from a well-established hard problem in lattice-based cryptography: Learning With Errors (LWE). Specifically, in the CKKS decryption form $c \cdot sk \ mod \ Q = m + e$, the noise term $e$, which is intentionally added and accumulates throughout the computation process, ensures that the plaintext cannot be recovered in polynomial time without access to the secret key. Since $e$ is typically much smaller than the encrypted message $m$, it has a negligible impact on the accuracy of the final output. As discussed in (Cheon et al., 2017), fundamental homomorphic operations such as addition, multiplication, and rotation are all designed to preserve computational security. Furthermore, according to the threat model outlined in Section 2.4, the client does not disclose the secret key to the server or any third party. All homomorphic operations used in FicGCN are covered by the security analysis in (Cheon et al., 2017). In the setup stage, we select HE parameters(Cheon et al.,

2018) to achieve 128-bit security—meaning any successful attack would require at least $2^{128}$ basic operations.Therefore, the above theoretical foundations and configuration ensure the overall security of the inference process in FicGCN.

## 3. Method

### 3.1. Overview

In this work, we propose FicGCN from the following three aspects as shown in Figure 2: **1)** The latency-aware methods for packing to efficiently utilize ciphertext slots. **2)** The novel SpIntra-CA algorithm, which utilizes graph sparsity to reduce the aggregation overhead. **3)** The Node Order Optimizing (NOO) algorithm further enhances the efficiency of SpIntra-CA with aggregation-friendly node arrangement.

### 3.2. Latency-Aware Packing

Since decryption is not possible during the entire inference process, the initial packing strategy is crucial. The packing scheme determines both the utilization of ciphertext slots and the order of nodes within the ciphertext.

Our primary packing principle is designed to be Combination-friendly since the Combination stage does not exhibit sparsity, which thus is suitable and efficient for SIMD computation. Let $M$ denote the number of ciphertext slots, $N$ the number of nodes, $F$ and $F'$ the feature dimensions before and after the layer computation, and $n$ the number of sampled neighbors. That is, given a feature matrix $X \in \mathbb{R}^{N \times F}$ we perform the packing on a column-wise basis as the blue ciphertexts in Figure 1(a).

However, due to the varying dimensions of $A$ and $X$ across different graphs, column-based packing may not always guarantee efficient utilization of ciphertext slots. When the column vector is of small dimension, the packing scheme that assigns one ciphertext per column results in a waste of slots and an excessive number of ciphertexts, leading to significantly high Homomorphic Operation Counts (HOCs).

To address the issue, we model the impact of $t$ (column number in one ciphertext) on latency and compute the latency-optimal packing parameters, as shown in Figure 2(b). We can deduce that (The detailed derivations of the following two cases are provided in the appendix):

**1) When $\mathbf{M} > \mathbf{N} * \mathbf{F}'$:** The objective function can be approximated as

$$\mathcal{J}(t; F, n) = 2 \lceil \frac{F * n}{t} \rceil + 20 \lceil \ log(t) \rceil$$

We can optimize the value of $t$ for this function and determine the packing strategy (shown in Appendix Section B).

**2) When M ≤ N ∗ F′:** It can be deduced that:

$$PMult\# = Add\# = \lceil \frac{F * n}{t} \rceil * \lceil \frac{N * F'}{\lceil \frac{M}{t} \rceil} \rceil$$

This is an approximation that is independent of $t$, meaning that the total latency only depends on the number of Rot. Therefore, the optimal solution is $t = 1$.

### 3.3. Sparse Intra-ciphertext Aggregation

In this section, we introduce the innovative Sparse Intra-ciphertext Aggregation algorithm, which effectively exploits the graph sparsity and improves the performance.

The existing HE+GCN aggregation method does not utilize the sparsity of GCN and thus is efficient (Ran et al., 2024). They construct $N$ ciphertexts, each containing duplicate nodes $node_i$, and perform aggregation using the adjacency matrix $A$. However, due to the sparsity of $A$, this method introduces significant redundant computation and storage.

Observing that GCN exhibits significant sparsity, aggregation only performs with neighboring nodes. Thus, by constructing $n$ "neighboring ciphertext", where each node position holds its neighboring node, we can achieve parallel aggregation across all nodes and eliminate the redundant computation.

However, Due to the irregular connectivity in GCN, neighboring nodes of each node are randomly arranged in the ciphertext, making it difficult to efficiently obtain the "neighbor ciphertexts". The naive method (Ru et al., 2021) as shown in Figure 3(a), performs node-by-node extraction, but results in an $O(N)$ rotation complexity, where each rotation is only effective for one node, leading to low efficiency.

Inspired by the ciphertext internal-sum method, which rotates each slot total $M - 1$ steps (where $M - 1 = 2^0 + 2^1 + ... + 2^{\log(M)-1}$), this method executes rotations sequentially (i.e., $2^0, 2^1, 2^2...$) and performs a reduce to achieve summation. This method allows each rotation to be effective for all slots, ultimately requiring only $\log(M)$ rotations, significantly reducing needed rotations. The process can be represented by the following expression:

$$ct \leftarrow ct \oplus Rot(ct, 2^m) \ , m = 0, 1, .., log(M) - 1$$

Based on this, we propose the SpIntra-CA method as shown in Figure 3 to efficiently generate neighbor ciphertexts and achieve efficient aggregation. **First**, given the ciphertext, we calculate the rotation length required for each node based on $A$, to construct the target "neighbor ciphertext" to be aggregated. Due to the irregular arrangement of sibling nodes in GCN, the rotation length varies for each node. **Second**, we perform bit decomposition of the rotation lengths for all nodes, executing rotations for all nodes sequentially from

*Table 1.* Ablation Study for AOO

| Model | Rot | PMult | Add | Latency(s) |
|---|---|---|---|---|
| Inter-CA | 0 | 189K | 206K | 70.08 |
| w/o AOO | 7.04±0.78K | 61±3.1K | 61±3.1K | 47.65±3.76 |
| w/ AOO | 5.74±0.25K | 65±4.0K | 65±4.0K | 40.39±1.61 |

the lowest bit. Let the $i-$th rotation step be $bit_i^j * 2^i$ for $node_j$, no rotation is performed (retained in the current slot) when $bit_i^j$ is equal to 0. In an ideal scenario (where all $bit_i = 1$), the $i$-th rotation would be effective for all nodes, improving the rotation efficiency significantly and allowing us to obtain the neighbor ciphertext in at most $\log(M)$ rotations. **Meanwhile**, we also check whether each node has reached its target position; if it has, it is removed from the current ciphertext and added to the result ciphertext.

All of the removal and position-selecting operations mentioned above can be accomplished by multiplying the ciphertext with a "Mask" plaintext polynomial composed of 0 and 1. Our SpIntra-CA can be iteratively represented as follows:

$$\{ct\} \leftarrow \{ct\} \otimes Mask_1 \oplus Rot(\{ct\}, 2^{m-1}) \otimes Mask_2$$

#### 3.3.1. REDUCING COMPUTATIONAL COMPLEXITY

As shown in Figure 3(d), SpIntra-CA experiences slot conflicts when multiple nodes attempt to occupy the same slot. This results in additional ciphertexts to resolve these conflicts and more extra rotation overhead. Therefore, measures must be implemented to either reduce conflicts or minimize the shifting range of each node within the ciphertext.

**Worst Case Analysis**: In the worst case, each node traverses at most $log(N)$ slots and collides at all of them, leading to at most $log(N)$ extra ciphertexts and $log^2(N)$ Rots at all. Further, considering the number of ciphertexts (sampled neighbors), the total number of rotations is $(n-1)*log^2(N)$. In practice, conflicts occur less frequently than assumed. The results in Section 4 will demonstrate that, even when using the SpIntra-CA algorithm without reordering, the Rot number is often much lower than the worst-case scenario.

**Optimizations:** To improve the efficiency of SpIntra-CA, we have proposed the following optimizations:

- **Aggregation Order Optimization (AOO):** As shown in Figure 4(a), observing the cyclic shift property of ciphertext rotation, we have optimized the shift order of each node to find the shortest shift distance. This can reduce the bits of each node's shift step of SpIntra-CA, thereby reducing the Rot number.

- **Ciphertext Processing Order Optimization (CPOO):** As shown in Figure 3(b), both the conflict

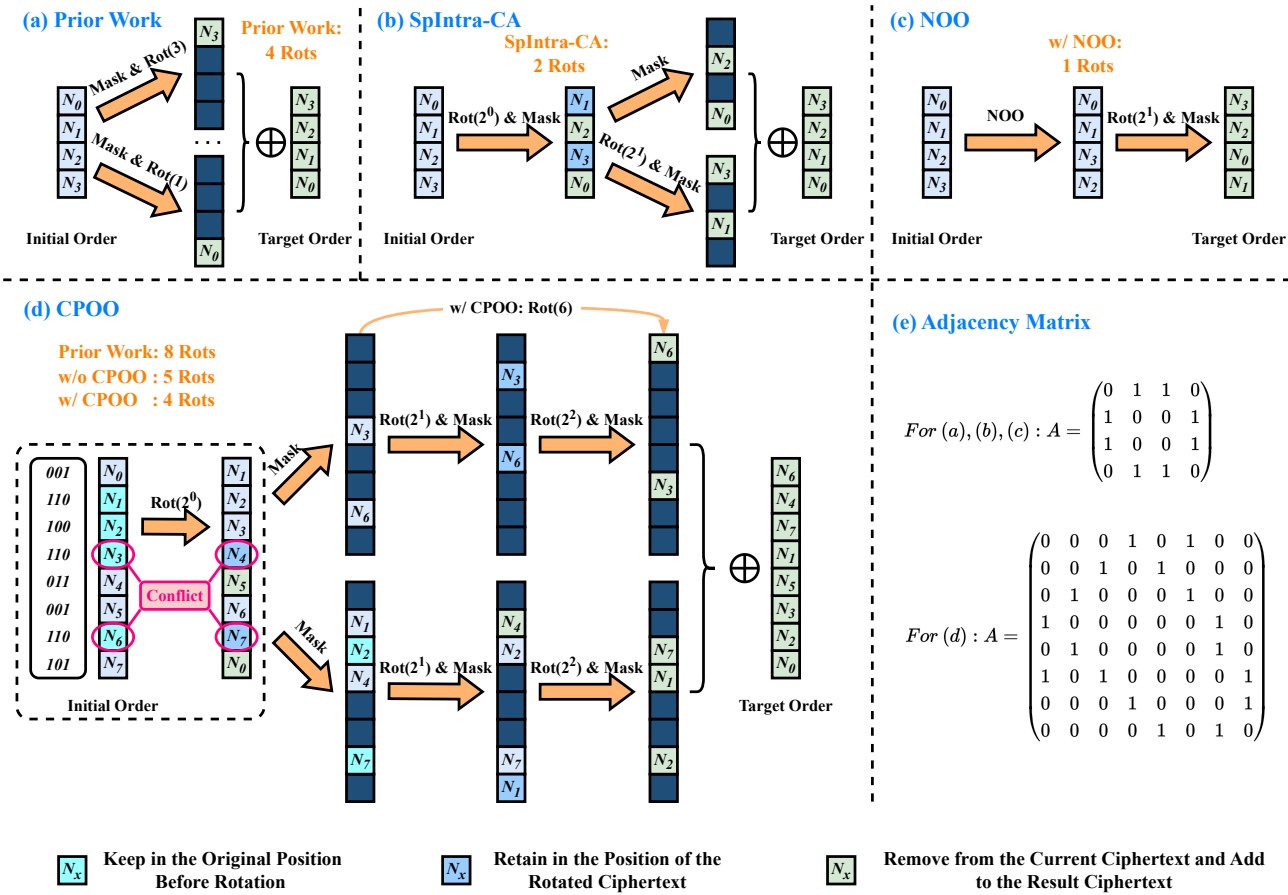

*Figure 3.* A toy example of SpIntra-CA. (a) Aggregation in prior work node by node. (b) The process of SpIntra-CA. (c) The NOO effect. (d) Conflicts arise when multiple nodes occupy the same slot, and removals occur when a node reaches its aggregation target position. Also CPOO may help merge some common Rots in sparse ciphertexts

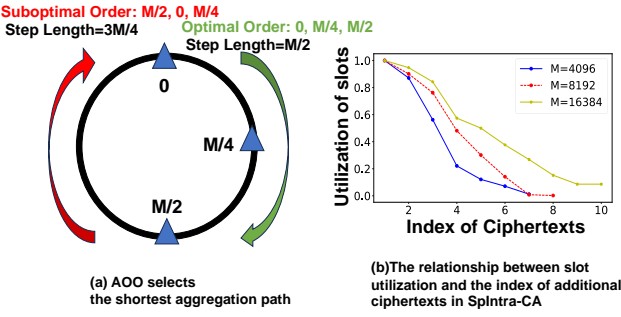

*Figure 4.* Aggregation and ciphertext processing order problem in SpIntra-CA

and removal in SpIntra-CA lead to more sparse ciphertext, which decreases the efficiency of SIMD. Observing that no conflicts occur in continuous multiple rotations on sparse ciphertext, we propose to merge them into one large-step rotation to reduce the

rotation counts and increase the node number per Rot. For example, we merge Rot(2) and Rot(4) into Rot(6) in Figure 3(d). Meanwhile there exists oppurtunity to merge sparse ciphertexts into a denser one, which is another part of CPOO.

- **Node Order Optimization (NOO):** The order of nodes within the ciphertext is also crucial as illustrated in Figure 3(c). This will be elaborated in detail in Section 3.4.

### 3.3.2. SELECTING AGGREGATION MODE

In FicGCN, two Aggregation modes are available: SpIntra-CA and Inter-CA. For each layer, the optimal mode is selected based on complexity analysis. From Table 2, we can deduce that the computational complexity of Inter-CA primarily arises from multiple inter-ciphertext calculations to obtain different node arrangements, which is mainly determined by $F$. In contrast, SpIntra-CA involves rotating a single ciphertext to generate others, and is mainly influenced

*Table 2.* The comparison of HOC for different Aggregation modes. '/' indicates that HOC for this mode cannot be theoretically analyzed.

| | | Complexity | |
|---|---|---|---|
| Operation | Latency | SpIntra-CA | Inter-CA |
| PMult | Low | $\ll N$ | $\lceil \frac{F*n}{t} \rceil$ |
| Add | Low | $\ll N$ | $\lceil \frac{F*n}{t} \rceil$ |
| Rot | High | $O(n*log^2(N))$ | $\lceil log(t) \rceil$ |

by $N$.

Based on this, the scheduling principle for layer-wise Aggregation mode can be derived as follows:

- **First Layer**: We always choose Inter-CA for the first layer, as the multiple inter-ciphertext computations can be eliminated by offline user encryptions from the client, which can be ignored.

- **Other Layers**: For other layers, considering that Rot is often more than $20\times$ slower than Pmult and Add (Lee et al., 2022), we follow the latency-aware formula (Table 2) to select Aggregation mode. In SpIntra-CA, we introduce a factor $c \in [0, 1]$ to quantify the effect of the optimization (like AOO) and the actual Rot number in practice (that is often less than the worst-case scenario). The final formula is as follows:

$$Agg = \arg\min \begin{cases} \text{Inter-CA:} & 2\lceil \frac{F*n}{t} \rceil \\ \text{SpIntra-CA:} & 10cn\log^2(N) \end{cases}$$

SpIntra-CA is preferred when $F$ dominates the latency.

### 3.4. Node Order Optimization

As analyzed in Section 3.3.1, the efficiency of SpIntra-CA is limited by the maximum rotation range and the conflict number. The large rotation range results in a longer rotation step, leading to more rotations; Excessive conflicts may lead to more extra ciphertexts and rotations. From Section 3.3, we can observe that the rotation step is closely tied to the relative distance between the nodes to be aggregated in the ciphertext, which aligns seamlessly with the regional characteristics observed in the graph. Moreover, the arrangement of nodes within a region plays a crucial role in determining the frequency of conflicts.

Based on the observations above, FicGCN proposes node order optimizations (NOO) for HE+GCN to reduce HOCs and enhance efficiency, which includes the following three stages: **1) BFS-based Region Partition**. We sequentially obtain all regions and find the nodes for each region with Breadth-First Search (BFS). The only adjustment is that

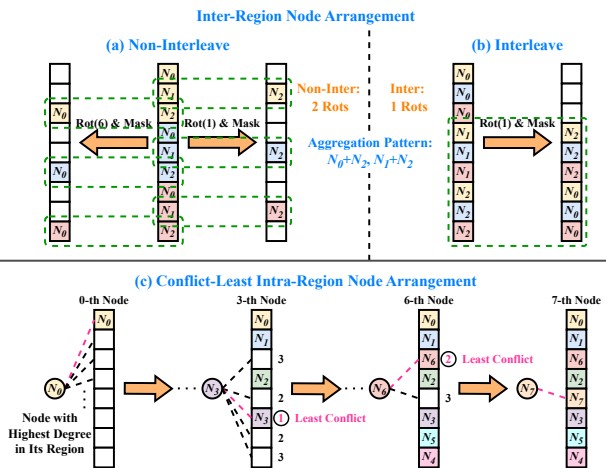

*Figure 5.* Node order optimization (NOO). Interleaved region arrangement (b) requires fewer rotations than the region-by-region arrangement (a). Greedy node arrangement for least conflicts within each region (c)

we enqueued sibling nodes instead of neighboring nodes in each iteration. We also use a threshold $TH$ to limit the node number within a region. **2) Aggregation-efficient inter-region node arrangement**: Aggregation occurs between sibling nodes, primarily within the same region. We propose an interleaved arrangement of nodes between regions where each region can be regarded as a subring of the ciphertext, preserving the rotation pattern as illustrated in Figure 5(b). This inherent consistency further facilitates parallel computation across regions. **3) Conflict-least intra-region node arrangement**. Within each region, we employ a greedy strategy to sequentially determine node positions by selecting the $(k + 1)$-th node that minimizes the total number of conflicts with the already fixed $k$ nodes, as shown in Figure 5(c). Thanks to the interleave arrangement, the search within each region can be conducted independently without interfering with one another.

Existing work also designed reordering algorithms, which are designed to enhance cache hit rates for plaintext computing (Geng et al., 2021; Wei et al., 2016; Arai et al., 2016). It fails to achieve optimal performance because it is not designed to accommodate the ring structure of CKKS ciphertexts and cannot effectively reduce conflicts within a region. The NOO Algorithm is shown in Appendix.

## 4. Evaluation

### 4.1. Experiment Setup

**Datasets.** We conduct experiments on the four datasets the same as the existing works (Ran et al., 2022; 2024), including Cora, Citeseer, Corafull and NTU-cross-View. The former three are relatively large-scale graphs, comprising 2708, 3327, and 19793 nodes with feature dimensions of

1433, 3703, and 8710, respectively. Notably, the Corafull dataset exceeds the complexity of all prior inferred datasets in HE-GCN. The last one is a small-scale dataset designed for human action recognition, comprising 25 nodes with 3-dimensional features. However, due to its inclusion of 256 frames with a temporally dense computation pattern, the computational latency under HE remains high.

**Models.** Graph Auto-Encoder (GAE) models with 2 layers and 2 non-linear activation functions(approximated by $x^2$) are trained on three large-scale datasets, using Adam optimizer with a mini-batch size of 64, a momentum of 0.9, and the learning rate of 0.01 for 200 epochs. The feature dimensions of the 2 layers are set to 32 and 16, respectively. We employ a number of sampled neighbors that is approximately equal to the average number of neighbors per node in the original dataset. We also set $TH = 1024$ for Cora and Citeseer; $TH = 4096$ for Corafull. For NTU-cross-View, we train using the identical STGCN-3-64 model (Ran et al., 2022) and hyper-parameters following CryptoGCN.

**HE parameters & Environment.** For the datasets with large and small scales described above, we respectively adopt the HE parameter configurations from Penguin (Ran et al., 2024) and CryptoGCN (Ran et al., 2022). For large-scale datasets, we set: $\Delta = 2^{30}$ ; $M = 2^{12}$ ; $Q = 218$. For small-scale datasets, we set: $\Delta = 2^{33}$ ; $M = 2^{13}$ ; $Q = 680$. We conduct all experiments on a machine equipped with Intel(R) Core(TM) i7-9750H CPU using the single thread setting to test the inference latency and use Microsoft SEAL version 3.7.2 (SEAL)to implement the CKKS scheme.

## 4.2. Evaluation Results

### 4.2.1. COMPARE WITH SOTA SOLUTIONS

We conduct an end-to-end latency comparison analysis with SOTA methods including Gazelle, Penguin, and CryptoGCN. For a fair comparison, when using Penguin, we perform aggregation with the plaintext adjacency matrix $A$. As shown in Table 3, our FicGCN achieves speedups of $21.6\times$, $34.2\times$, $>120\times$ over Gazelle across Cora, Citeseer, and Corafull, respectively. The results in Table 3 also show that on the NTU dataset (25 points), we achieve a $1.26\times$ speedup over the fastest design, CryptoGCN. On the Cora (2708 points) and Citeseer (3327 points) datasets, we achieve $2.01\times$ and $1.78\times$ speedups, respectively, compared to the fastest design, Penguin. Furthermore, on the Corafull dataset (19793 points), our method outperforms the fastest design CryptoGCN, by a factor of $4.1\times$.

This comparison highlights that our method delivers stronger performance on larger graphs. Specifically, SpIntra-CA enables arbitrary reordering within the ciphertext, effectively addressing the inefficiency that arises when $M \ll N$. In contrast, existing approaches such as Penguin face per-

*Table 3.* Compare with SOTA solutions. (*"E" represents Inter-CA, "A" represents SpIntra-CA, and "Hybrid" denotes a mixed computation of the two methods.*)

| Dataset | Method | Aggregation | Slot Utilization | Latency | Speed up |
|---|---|---|---|---|---|
| Cora | Gazelle | - | 35% | 1535.27s | - |
| | Penguin | Hybrid | 100% | 128.82s | 11.9× |
| | CryptoGCN | E-E | 66% | 131.06s | 11.7× |
| | **FicGCN+NOO** | E-A | 100% | **64.12s** | **21.6×** |
| Citeseer | Gazelle | - | 81% | 2897.25s | - |
| | Penguin | Hybrid | 100% | 142.90s | 20.3× |
| | CryptoGCN | E-E | 90% | 150.42s | 19.3× |
| | **FicGCN+NOO** | E-A | 100% | **79.98s** | **36.2×** |
| Corafull | Gazelle | - | 72% | / | - |
| | Penguin | Hybrid | 100% | 35565s | >30× |
| | CryptoGCN | E-E | 83% | 31735s | >30× |
| | **FicGCN+NOO** | E-A | 100% | **7733s** | **>120×** |
| NTU | CryptoGCN | E-E-E | 78% | 1731.08s | - |
| | **FicGCN** | E-E-A | 100% | 1373.82s | **1.26×** |

formance bottlenecks under these conditions, limiting their overall efficiency. We outperform state-of-the-art (SOTA) methods primarily due to two factors:(1). Our SpIntra-CA overcomes the trade-off between sparsity and SIMD packing, significantly reducing redundant computations. While Penguin uses efficient packing but not sparsity, and CryptoGCN adopts both sparsity and SIMD but suffers low slot utilization, our approach is more holistic. (2). We devise a specialized reordering algorithm for SpIntra-CA and CKKS's cyclic shift, which trades minimal plaintext overhead for a significant reduction in homomorphic latency.

### 4.2.2. ABLATION STUDY

**Ablation study for AOO.** We first conduct an ablation study for AOO. As illustrated in Figure 4(a), we determine the optimal aggregation order for each node to minimize the shift range within the ciphertext. Clearly, AOO is independent of the dataset, thus, without loss of generality, we analyze this only using the Cora dataset as an example. We report the mean and standard deviation of results from 10 different samples.

Table 1 shows that incorporating AOO results in an 18.5% rotation reduction and achieves a speedup of 1.18×. AOO may introduce minor side effects since multiple nodes may occupy the same optimized initial position, which may incur slightly more conflicts in SpIntra-CA, but still leads to the above performance gains.

**Ablation Study for CPOO.** As shown in Figure 6(b), CPOO that merging and postponing the process of sparser ciphertexts enhances computational efficiency by 15%, pruning 14%-46% Rots, which is attributed to allowing a single rotation to satisfy the requirements of more nodes with a relatively low conflict rate. Figure 6(a) illustrates the ratio of inference latency under different ciphertext sparsity thresholds to that without CPOO across four datasets, which demonstrates that the optimal threshold should be selected

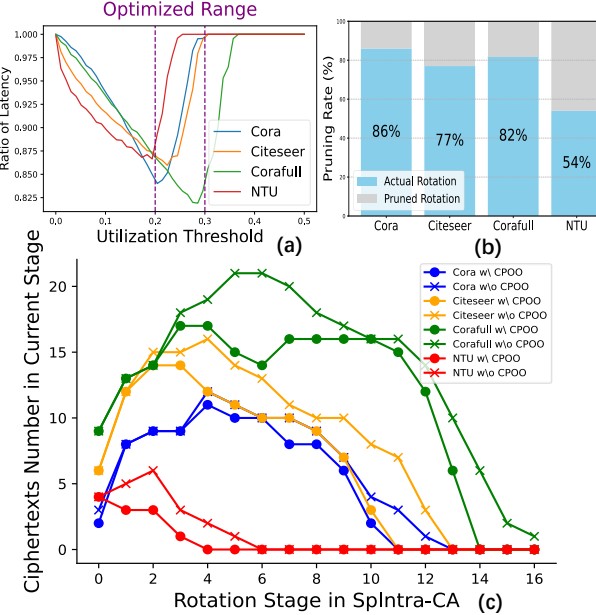

*Figure 6.* The effectiveness of CPOO. (a) The relationship between latency and utilization thresholds across four datasets. (b) The ratio of Rots pruned by CPOO of 4 datasets. (c) Stage-wise ciphertext number w/ and w/o CPOO

*Table 4.* Ablation Study for NOO (*NOR: Number of Regions*)

| Dataset | Model | Rot | PMult | Add | Latency(s) | Accuracy (Backbone) | NOR |
|---------|-------|-----|-------|-----|-----------|---------------------|-----|
| Cora | w/o NOO | 5.74K | 125K | 117K | 94.68 | 0.792 (0.815) | 6 |
| | w/ Rabbit | 2.85K | 109K | 103K | 80.05 | | |
| | w/ NOO | 1.93K | 97K | 95K | 69.28 | | |
| Citeseer | w/o NOO | 7.03K | 151K | 139K | 117.87 | 0.692 (0.727) | 8 |
| | w/ Rabbit | 3.60K | 140K | 131K | 97.21 | | |
| | w/ NOO | 2.35K | 123K | 120K | 86.14 | | |
| Corafull | w/o NOO | 46.3K | 20.0M | 22.8M | 12003 | 0.648 (0.695) | 14 |
| | w/ Rabbit | 38.3K | 16.5M | 17.5M | 10011 | | |
| | w/ NOO | 36.7K | 14.7M | 16.9M | 9075 | | |
| NTU | w/o NOO | 10.66K | 233K | 257K | 1562.03 | 0.749 (0.825) | 2 |
| | w/ Rabbit | 10.66K | 233K | 257K | 1562.03 | | |
| | w/ NOO | 9.97K | 218K | 241K | 1463.80 | | |

within 0.2~0.3. Figure 6(c) further illustrates the effectiveness of CPOO across different datasets.

**Ablation Study for NOO.** NOO performs a greedy search for the node with the fewest current conflicts in SpIntra-CA, resulting in optimization performance that surpasses existing cache hit rate improvement methods such as Rabbit (Arai et al., 2016). Table 4 presents the ablation experiments of NOO on 4 datasets. Our proposed NOO reduces Rot number by 66.4%, 66.6%, 20.7% and 6.5% compared with direct SpIntra-CA respectively. The relatively lower improvement in NTU-cross-view dataset is attributed to its smaller node number (25) and simpler connectivity structures, in which cases we suggest not using NOO to save offline time. Among the 3 optimization methods, NOO demonstrates the most significant effect.

*Table 5.* Overhead analysis of Node Order Optimization (NOO) on large-scale datasets

| Dataset | $|V(G)|$ | $|E(G)|$ | $d_{avg}$ | $T_{NOO}$ (s) | $T_{HE}$ (s) | $\rho$ |
|---------|----------|----------|-----------|---------------|--------------|--------|
| Cora | 2.7K | 5.4K | 4.01 | 2.00 | 64.12 | 3.2% |
| Citeseer | 3.3K | 4.7K | 2.85 | 5.33 | 79.98 | 6.7% |
| Corafull | 19.8K | 127K | 12.82 | 27.91 | 7733 | 0.36% |
| Pokec | 1.63M | 30.62M | 18.80 | 181.20 | $\sim 10^7$ | $\sim 0.002\%$ |

### 4.3. NOO Overhead Analysis

As shown in the results of Table 3, the advantages of FicGCN become more pronounced on large-scale graph datasets. The datasets used in current mainstream studies and in the evaluation of homomorphic inference in FicGCN all contain fewer than 100K nodes. Consequently, as the dataset size increases, the overhead of pre-processing NOO may potentially surpass online inference latency and become the primary performance bottleneck for FicGCN. To investigate this, we selected Pokec, a large-scale dataset with over 1 million nodes, to measure the time overhead of NOO and compare it with the online inference latency. The results are presented in Table 4, which includes:

- Graph Statistics: Node count ($|V(G)|$), edge count ($|E(G)|$), and average degree ($d_{avg}$).

- Time Overhead: NOO preprocessing time ($T_{NOO}$) and online stage HE computation latency ($T_{HE}$).

- Efficiency Ratio: $\rho = T_{NOO}/T_{HE}$

As shown in Table 5, NOO exhibits low time overhead on smaller datasets. For large-scale graphs like Pokec, NOO's overhead remains negligible relative to FicGCN's online phase, ensuring NOO does not bottleneck FicGCN's performance.

## 5. Conclusion

In this paper, we explore the performance potential of sparsity in HE+GCN. Based on the sparsity of node connection, we propose FicGCN with an optimal layer-wise column-based packing method and a sparse intra-ciphertext aggregation method to reduce the computation redundancy and boost the performance. We also propose a region-based data reordering method to further improve the aggregation efficiency. The results show an up to 4.1x speedup compared to SOTA works and demonstrate its superiority on large graphs, promoting the practical deployment of HE GCN.

## Acknowledgments

This work is partially supported by Strategic Priority Research Program of the Chinese Academy of Sciences, (Grant

No.XDB0660300, XDB0660301, XDB0660302), the NSF of China(under Grants U22A2028, 62222214, 62341411, 62102398, 62102399, 62302478, 62302482, 62302483, 62302480, 62302481, 62172387), CAS Project for Young Scientists in Basic Research(YSBR-029) and Youth Innovation Promotion Association CAS.

## Impact Statement

Our paper enhances the computing efficiency of Homomorphic Encryption (HE) GCNs while ensuring data privacy through robust security guarantees of HE. With the widespread use of GCN systems and growing privacy concerns, our paper facilitates the practical application of GCNs in security and privacy-critical scenarios. We believe our work positively impacts society.

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

# A. HE Operation Overhead Analysis

As mentioned in the main text, the basic overhead of different types of HE operations varies. Generally, $L(Rot) \approx L(CMult) > L(PMult) > L(Add)$, 'L' represents latency. The two aggregation modes primarily used in this paper (Intra-CA and Inter-CA) rely on different fundamental operations. By analyzing the differing HOCs for each mode, it is evident that each method has its strengths and weaknesses depending on the dimension of the feature matrix, the size of the model, and the HE parameters configuration.

Since CKKS is a leveled homomorphic encryption (LHE) scheme, it divides the bits of each coefficient in the ciphertext polynomial into multiple groups, with each group corresponding to a multiplication level. After performing a multiplication on the ciphertext, a re-scaling operation is necessary to reduce the bit length of the coefficients, lowering the cost for subsequent HE operations. The core idea behind an optimal inference framework is to prioritize the cheaper operations, such as Add, PMult, and delay the more expensive ones. This is reflected in FicGCN's design, where the first layer primarily uses the Inter-CA based on PMult and Add, while the SpIntra-CA is placed in later layers.

Figure 7(a) illustrates the HE operation individual cost at different levels for ciphertexts under the HE parameter configuration used for large graph inference in this paper. It shows that the overhead of Rot and CMult is often over $20\times$ greater than that of PMult and Add. Figure 7(b) breaks down the time overhead of HE operations under different methods. In inference dominated by SpIntra-CA, the delay caused by Rot exceeds A%. Therefore, the methods proposed in this paper, such as AOO, CPOO, and NOO, which reduce the overhead of Rot to enhance computational efficiency, have been shown to be effective. Since under plaintext $A$, the computational overhead of the non-linear layer represented by CMult is negligible, there is no need to prune the non-linear layer as done in prior work (Lee et al., 2023; Peng et al., 2024; Ao & Boddeti, 2024; Jha et al., 2021; Ran et al., 2023).

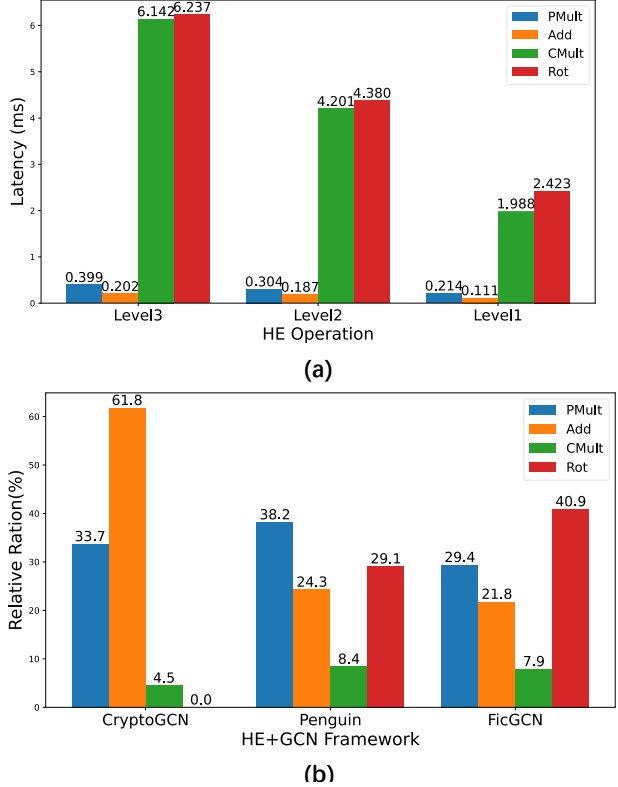

*Figure 7.* (a) Statistics of single operation overhead for the four basic HE operations at different levels of ciphertext. (b)Breakdown of time overhead for the 4 basic HE operations under different inference frameworks.

# B. Latency-Aware Objective Function

Section 3.2 presents two macro scheduling strategies for FicGCN—packing method and layer-wise aggregation mode decisions. The commonality of these two strategies is that they estimate the latency by evaluating the HOC under different methods, which is then used as the objective function to derive the optimal solution and strategy for the current state. This section provides a detailed derivation of how the two objective functions are obtained.

*Table 6.* Experiments Setup

| Dataset | Model | Feature Dimension | Average Degree (Sampled Neighbors) | HE Parameters | | | Security Level |
|---------|-------|-------------------|-----------------------------------|---|---|---|----------------|
| | | | | M | P | Q | |
| Cora | GCN | 1433-32-16 | 4.01 | 4096 | 30 | 218 | 128-bit |
| | GraphSage | | 4.00 | | | | |
| Citeseer | GCN | 3703-32-16 | 2.85 | 4096 | 30 | 218 | 128-bit |
| | GraphSage | | 3.00 | | | | |
| Corafull | GCN | 8710-32-16 | 12.82 | 4096 | 30 | 218 | 128-bit |
| | GraphSage | | 13.00 | | | | |
| NTU | GCN | 3-64-128-128 | 2.00-4.00 | 8192 | 60 | 680 | $\geq$ 80-bit |
| | GraphSage | | 4.00 | | | | |

## B.1. Packing Method

As shown in Figure 1(a), to minimize the dense computation overhead in the Combination step, we propose a column-based packing strategy. Specifically, we pack the data from $t$ columns of $X$ into the same ciphertext polynomial in column-major order. When $t = 1$, each ciphertext stores a single feature dimension for all nodes, eliminating the need for expensive rotation operations during the Combination phase. However, since the dimensions of $X$ vary, $t = 1$ may not guarantee maximum ciphertext slot utilization, potentially resulting in more ciphertexts and additional overhead. Therefore, we must balance ciphertext utilization with the extra rotation overhead, aiming to find the optimal value of $t$ by using the first-layer Inter-CA latency estimation as the objective function.

We divide the problem into two categories based on whether the ciphertext size can fully pack all the elements of the next layer's feature matrix, i.e. by comparing the sizes of $M$ and $N * F^{'}$.

$\mathbf{M} > \mathbf{N} * \mathbf{F}^{'}$:In this case, it is possible to pack a particular column from Figure 1(a) within the same ciphertext, so the number of PMult and Add is $\lceil \frac{F*n}{t} \rceil$, where $\lceil \cdot \rceil$ denotes ceiling. Since there are multiple feature dimensions within the ciphertext, considering the intra-ciphertext summation , this will incur an additional rotation overhead of $log(t)$ times. (Halevi & Shoup, 2014) Here we consider $L(CMult) = L(Rot) = 20L(PMult) = 20L(Add)$, thus we can derive the objective function as:

$$\mathcal{J}(t; F, n) = 2 \lceil \frac{F*n}{t} \rceil + 20 \lceil log(t) \rceil$$

Figure 7(a) shows an example objective function, the optimal $t$ can be searched within a cheap pre-processing overhead. $\mathbf{M} \leq \mathbf{N} * \mathbf{F}^{'}$: In this case, we are unable to fully pack a complete column from Figure 1(a) into a single ciphertext; we can only pack several rows. Since we pack $t$ columns from $X$ each time, a single ciphertext will contain $\lceil \frac{M}{t} \rceil$ rows as shown in the figure. Consider that the matrix in Figure 1(a) contains $N * F^{'}$ rows, we can get $\lceil \frac{N*F^{'}}{\lceil \frac{M}{t} \rceil} \rceil$ rows in the figure within a single ciphertext. Therefore, the number of PMult and Add can be calculated as:

$$Num(PMult) = Num(Add) = \lceil \frac{F*n}{t} \rceil * \lceil \frac{N*F^{'}}{\lceil \frac{M}{t} \rceil} \rceil$$

This is an approximation that is independent of $t$, meaning that the total latency only depends on the number of Rot. Therefore, the optimal solution is $t = 1$.

### B.2. Selecting Aggregation Mode

From Table 1, it can be seen that the dominant HE operations in Inter-CA and SpIntra-CA differ. Inter-CA primarily involves $n$ different orderings of repeated ciphertext-level additions and multiplications, so the time overhead can be estimated as: $2\lceil \frac{F*n}{t} \rceil$. On the other hand, SpIntra-CA is mainly composed of a single-order ciphertext, from which a new permutation of ciphertext is generated every $log^2(N)$ rotations. As a result, a smaller time overhead can be chosen for each layer, and the aggregation mode for each layer can be determined before starting the inference.

$$Agg = \arg\min \begin{cases} \text{Inter-CA:} & 2\lceil \frac{F*n}{t} \rceil \\ \text{SpIntra-CA:} & 10cn\log^2(N) \end{cases}$$

At the same time, since the final output and the node order in the client's packed ciphertext can be arbitrary, the NOO results on the SpIntra-CA results can be propagated back to the client layer by layer to complete the optimal order packing since there is no restriction for both the order of the input and output in FicGCN, as shown in Figure 7(b).

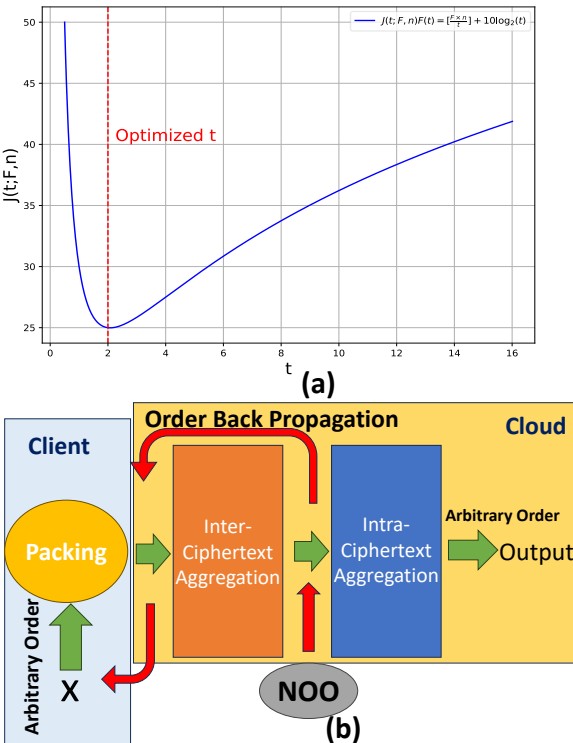

*Figure 8.* (a) An example objective function (b) The principal of node order back propagation which influences the initial packing order from the client.

## C. Supplementary Experiments

### C.1. Experiments Setup

All experimental settings are mentioned in Section 4.1. Here, this information is summarized in Table 5. When sampling, the number of neighbors we sample is approximately equal to the average degree of all nodes in the graph.

### C.2. Ablation Study for Packing

We pack $t$ columns from $X$ into a ciphertext and use the delay estimate of the first-layer Inter-CA as the objective function to find the optimal t value, balancing the utilization of ciphertext slots and the extra Rot overhead induced. However, whether this approach remains effective for subsequent layers, particularly for those using SpIntra-CA, is a nontrivial question.

*Table 7.* Ablation Study for Packing

| DataSet | Optimized $t$ | Packing Method | Ciphertext Number | Inter-CA Latency | SpIntra-CA Latency | Overall Latency | Speedup with Optimized $t$ |
|---------|---------------|----------------|-------------------|------------------|--------------------|-----------------|----------------------------|
| Cora | 2 | $t = 1$ | 1433-32-16 | 37.95s | 8.07s | 46.02s | |
| | | $\mathbf{t = 2}$ | 717-16-8 | 20.36s | 15.87s | **36.23s** | 1.27× |
| | | $t = 4$ | 359-8-4 | 14.88s | 27.01s | 41.89s | |
| Citeseer | 2 | $t = 1$ | 3703-32-16 | 77.00s | 5.98s | 82.98s | |
| | | $\mathbf{t = 2}$ | 1852-16-8 | 40.29s | 17.47s | **57.76s** | 1.44× |
| | | $t = 4$ | 926-8-4 | 21.69s | 39.05s | 60.74s | |

For Inter-CA, since the ciphertexts involved in the computation are repeatedly generated, the overall computational cost is almost proportional to the number of ciphertexts. Therefore, reducing the number of ciphertexts, or in other words, increasing the packing density, can reduce the latency. However, for SpIntra-CA, the situation is more complicated. As the slots utilization increases, the sparsity of ciphertexts decreases significantly, resulting in a higher probability of conflicts during computation, which actually increases the latency. Furthermore, since these conflicts are closely related to the "Graph Coloring Problem", which is NPC, the exact relationship between conflict occurrences and ciphertext sparsity is hard to be theoretically derived.

Therefore, we conduct experiments to verify the effectiveness of the proposed latency-aware packing optimization method. Table 6 shows the latency variation of different parts of the model under different packing strategies. Each data point is the average of 20 different random samples, and only the Cora and Citeseer datasets, with moderate dimensions, were selected for the experiment. Furthermore, 1024 nodes were randomly sampled from each dataset for the experiments.

From the table, we can observe that for both datasets, when $t$ is doubled, it leads to a halving of the number of ciphertexts, along with a small increase in the Rot overhead. As a result, the Inter-CA delay approximately becomes half of the original value. However, the delay for SpIntra-CA does not exhibit a clear pattern as ciphertexts become more densely packed. Therefore, when considering both, the optimal computation efficiency is typically achieved at the point that minimizes the overhead for the first layer. Thus, the proposed packing strategy not only minimizes the Inter-CA overhead in the first layer but also ensures minimal delay even when different modes of SpIntra-CA are considered for subsequent layers.

## C.3. NOO Effect

The principle behind our proposed NOO that enhances the computational performance of homomorphic inference mainly involves two key aspects:

- GCN sibling node computation mode: By grouping nodes with computational relationships into the same region and mapping this relationship to the ciphertext's ordering, we ensure that nodes that need to be computed are placed closer together. This effectively reduces the displacement distance of all nodes within the ciphertext.

- Heuristic search with a greedy strategy: By searching for the state with the fewest conflicts, the entire aggregation process sees a significant reduction in the number of conflicts.

Algorithm 1 aligns with the 3 stages illustrated in Figure 2(a). Lines 1–11 describe the detection of strongly connected regions using a BFS-based approach. In each iteration, the sibling nodes of the current node are enqueued, and a threshold $TH$ is employed to control the maximum number of nodes per region. Subsequently, line 12 interleaves the nodes of these regions within the vector. These two steps also facilitate the partitioning of the problem into sub-problems, ensuring the search space is partitioned from the entire graph into multiple regions and thereby achieving the divide-and-conquer strategy. Finally, lines 13–27 employ a greedy search strategy to traverse the node within each region individually, placing each node in a position that minimizes the total number of conflicts in the current state. Here, $Con_j^{rot}(pt_{Result}^r, n, i)$ indicates whether a conflict occurs at the $j - th$ position during the $rot - th$ rotation when node $n$ is placed in the $i - th$ position under the current node ordering. Moreover, the arrangement between these regions remains unaffected due to the SIMD rotational characteristics on the ring, thus the process of merging these sub-problems incurs no additional overhead.

As shown in Table 7, our NOO outperforms plaintext reordering algorithms for the following reasons: **1)** Most plaintext reordering algorithms place neighboring nodes closer together, but this doesn't directly align with the sibling computation patterns in GCN. **2)** Plaintext reordering algorithms are often designed to optimize cache hits in linear structures, while NOO focuses on the ring structure of CKKS ciphertext. **3)** Leveraging the computational features of CKKS, a divide-and-conquer

*Table 8.* Ablation Study for NOO

| Dataset | Model | Rot | PMult | Add | Latency(s) | Accuracy (Backbone) | NOR | Distribution of Ciphertext Number |
|---------|-------|-----|-------|-----|-----------|---------------------|-----|-----------------------------------|
| Cora | w/o NOO | 5.74K | 125K | 117K | 94.68 | 0.792 (0.815) | 6 | 2-8-9-9-11-10-10-8-8-6-2 |
| | w/ Rabbit | 2.85K | 109K | 103K | 80.05 | | | 3-7-8-7-6-5-4-3-1 |
| | w/ NOO | 1.73K | 91K | 89K | 64.12 | | | 2-5-5-5-4-3-2 |
| Citeseer | w/o NOO | 7.03K | 151K | 139K | 117.87 | 0.692 (0.727) | 8 | 6-12-12-12-12-11-10-10-10-9-3 |
| | w/ Rabbit | 3.60K | 140K | 131K | 97.21 | | | 3-9-9-8-7-7-7-3-2-2 |
| | w/ NOO | 2.21K | 118K | 115K | 79.98 | | | 2-7-6-4-4-3-3-2-2-1 |
| Corafull | w/o NOO | 46.3K | 20.0M | 22.8M | 12003 | 0.648 (0.695) | 14 | 9-13-14-17-17-15-14-16-16-16-16-15-12-6 |
| | w/ Rabbit | 38.3K | 16.5M | 17.5M | 10011 | | | 6-9-14-13-13-13-11-9-8-7-6 |
| | w/ NOO | 32.1K | 12.7M | 14.9M | 7733 | | | 4-11-11-8-10-7-7-6-5-3 |
| NTU | w/o NOO | 10.66K | 233K | 257K | 1562.03 | 0.749 (0.825) | 2 | 4-3-3-1 |
| | w/ Rabbit | 10.66K | 233K | 257K | 1562.03 | | | 4-3-3-1 |
| | w/ NOO | 9.97K | 214K | 236K | 1373.82 | | | 3-3-2-1 |

algorithm can naturally be designed to parallelize the processing of all partitioned regions, thereby reducing the number of conflicts.

NOO consists of three steps: Detection, Interleave Arrangement, and Search. The effects of each step will be presented one by one.

*Table 9.* Ablation Study for Packing

| Dataset | Naive Lat | Lat w/ Detection | Lat w/ Search | Lat w/ Rabbit | Lat w/ NOO |
|---------|-----------|------------------|---------------|---------------|------------|
| Cora | 94.68s | 76.73s | 88.22s | 80.05s | **64.12s** |
| Citeseer | 117.87s | 95.01s | 108.96s | 97.21s | **79.98s** |
| Corafull | 12003s | 10805s | 11426s | 10011s | **7733s** |
| NTU | 1562.03 | 1481.13s | 1413.45s | 1562.03s | **1373.82s** |

**Detection & Search Effect** As shown in Table 8, for medium-sized datasets like Cora and Citeseer, our Detection outperforms graph partitioning algorithms on plaintext, as prioritizing the grouping of sibling nodes into the same region better aligns with the computational structure of GCN. Additionally, Detection alone is more effective than Search alone, generally reducing latency by up to 13%. This suggests that, for these datasets, the disordered connectivity in randomly arranged ciphertexts is the primary factor limiting HE computation. However, for larger datasets like Corafull, where connectivity is more complex, the sibling-first detection strategy results in certain connected regions being inadequately explored, making Detection 8.0% slower than Rabbit. In such cases, Search is more effective, indicating that the key limiting factor for computational efficiency in large graphs is the higher probability of conflicts. For small-scale datasets like NTU, both Detection and Rabbit have little impact, with efficiency gains coming mainly from reducing conflicts.

Regardless of the dataset, **the efficiency improvement from using the complete NOO is always greater than the sum of the improvements from Detection and Search alone.** This is because only when both are used together can partitioned regions reduce the search space and enable parallel computation between regions (divide and conquer), resulting in higher computational efficiency.

**Interleave Arrangement Effect** We aim to arrange the nodes belonging to different regions in the ciphertext in an alternating pattern like ABCABCABC... as shown in Figure 2(a). However, in practice, we often encounter situations where the number of nodes in different regions is unequal, and the total number of nodes is slightly smaller than M. Therefore, we follow these principles for arrangement: 1) Placing the nodes of all in an interleaved manner. 2) When the nodes of a particular region are exhausted, prioritize filling the position of that region with blank slots. 3) If blank slots are also exhausted, begin alternately arranging the remaining regions. The final arrangement results for the four datasets are shown in Figure 8. We use gray to represent blank slots. As can be seen, each dataset ultimately faces a situation where the nodes of certain regions are exhausted, which breaks the efficient parallel rotation pattern established earlier. However, the proportion of nodes with this arrangement is quite low, so we can first complete the aggregation of most nodes and then handle this small portion of nodes at the end. This has little impact on the overall computational efficiency.

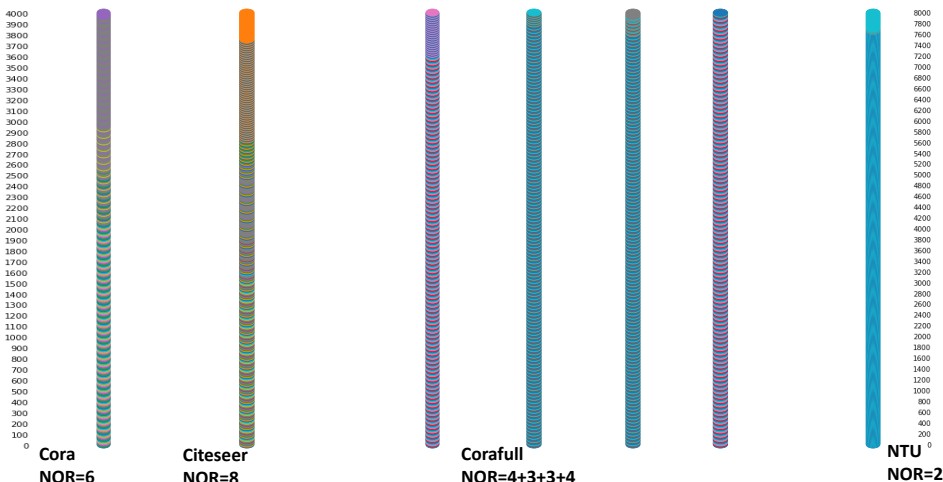

*Figure 9.* Interleave Arrangement Effect of 4 Datasets

---

**Algorithm 1** Node Order Optimization for SpIntra-CA

---

**Input:** Graph: $G = (V, E)$ ; $M$: Slot number; $TH$: Maximum number of nodes in one region; $\widetilde{A}$: Adjacency matrix after neighbor-sampling; $BFS_{sibling}$: A modified breadth-first search that enqueues the sibling nodes at each iteration.

**Output:** $pt_{Res}$: A plaintext vector which contains an optimized order of nodes

1: $\mathbf{R} \leftarrow \emptyset$ ; $Plain \leftarrow [\,]$ ; $V_{res} \leftarrow V$ ; $m \leftarrow 0$
2: **while** V is not empty **do**
3:     $N_{cur} \leftarrow Node\ with\ the\ highest\ degree\ in\ V_{cur}$
4:     $Num_{node} \leftarrow 0$; $R_{cur} \leftarrow \emptyset$
5:     $BFS_{sibling}.init(\widetilde{A}, N_{cur})$
6:     **while** $Num_{node} < TH$ or $BFS_{sibling}.Isend()$ **do**
7:        $N_{cur} \leftarrow BFS_{sibling}.pop()$
8:        $R_{cur} \leftarrow R_{cur} \cup N_{cur}$ ; $Num_{node} + = 1$
9:     **end while**
10:    $\mathbf{R}.push(R_{cur})$ ; $V \leftarrow V \backslash R_{cur}$
11: **end while**
12: $Plain \leftarrow InterleaveArrangement(\mathbf{R})$
13: $pt_{Res} \leftarrow Zeroslike(Plain)$
14: **for** $r = 1$ to $|\mathbf{R}|$ **do**
15:    $N'_{cur} \leftarrow Node\ with\ the\ highest\ degree\ in\ R_r$
16:    $pt_{Res}^r = Plain^r$ ; $pt_{Res}^r[1] \leftarrow N'_{cur}$
17:    **for** $n \in Sibling(N'_{cur})$ and $n$ is not visited **do**
18:       $k_{min} \leftarrow +\infty$ ; $index \leftarrow 0$ ; $L \leftarrow |Plain^r|$
19:       **for** $i = 2$ to $L$ **do**
20:          $k_n \leftarrow \sum_{j=1}^{L} \sum_{rot=0}^{log(L)-1} Con_j^{rot}(pt_{Res}^r, n, i)$
21:          **if** $pt_{Res}^r$ is available and $k_n < k_{min}$ **then**
22:             $index \leftarrow i$ ; $k_{min} \leftarrow k_n$
23:          **end if**
24:       **end for**
25:       $pt_{Res}^r[index] \leftarrow N'_{cur}$ ; $N'_{cur} \leftarrow n$
26:    **end for**
27: **end for**
28: **Return** $pt_{Res}$

---

