# OpenReview forum: "FicGCN: Unveiling the Homomorphic Encryption Efficiency from Irregular Graph Convolutional Networks"
_ICML.cc/2025/Conference — ICML 2025 poster_

### Official Review · Reviewer_JDUB · 2025-03-10

**Overall Recommendation:** 3

**Summary:**

This paper proposes FicGCN, a framework for efficient privacy-preserving inference of Graph Convolutional Networks (GCNs) using Homomorphic Encryption, by using (1) a latency-aware packing scheme that optimally balances aggregation and combination operations based on data dimensions and model structure; (2) a Sparse Intra-Ciphertext Aggregation method that minimizes rotation overhead in aggregation operations by leveraging graph sparsity; (3) a region-based node reordering technique that reduces computational overhead by optimizing the local adjacency structure.

**Claims And Evidence:**

The key claims appear to be well-supported by (1) detailed theoretical analysis and derivations of the optimization methods; (2) extensive experimental results across multiple datasets showing consistent improvements. However, a formal privacy analysis is missing for security properties claimed.

**Essential References Not Discussed:**

1. Zhang, Chengliang, et al. "{BatchCrypt}: Efficient homomorphic encryption for {Cross-Silo} federated learning." 2020 USENIX annual technical conference (USENIX ATC 20). 2020.
2. Chen, Tianyu, et al. "The-x: Privacy-preserving transformer inference with homomorphic encryption." arXiv preprint arXiv:2206.00216 (2022).
3.. Jin, Weizhao, et al. "FedML-HE: An efficient homomorphic-encryption-based privacy-preserving federated learning system." arXiv preprint arXiv:2303.10837 (2023).

**Experimental Designs Or Analyses:**

Experimental design seems to be sufficient. The analyses are generally valid, though they could benefit from additional privacy analysis, either from empirical attack simulation or statistical analysis like privacy sensitivity calculation.

**Methods And Evaluation Criteria:**

Same as above.

**Other Comments Or Suggestions:**

I would love to see a formal rigorous security proof. For example, use UC-Security to provide privacy analysis.

**Other Strengths And Weaknesses:**

Some technical details in appendix could be better integrated into main text to have a better flow for explanation of the proposed idea.

**Questions For Authors:**

1. How can the missing related work listed above further help justify the claims in the paper? Can any techniques in these papers be used to augment this work? For example, non-linear approximation and selective encryption?
2. How does the key management work in this work?
3. How does the system handle a client-server collusion threat model?

**Relation To Broader Scientific Literature:**

This work proposes a specific optimization when introducing HE into GCN for privacy-preserving ML. However, there still lacks some key related work that would raises concerns about the novelty and improvement claim in this paper. Some missing related work are listed below.

**Theoretical Claims:**

Detailed theoretical analysis and derivations of the optimization methods, but a formal privacy analysis is missing for security properties claimed.

---

> ### Author Rebuttal · Authors · 2025-03-31
>
> **1. Security proof of FicGCN**
>
> Thanks for your comments. FicGCN fully adopts CryptoGCN's[3] threat model and privacy assumptions which has been rigorously proved. We utilize the CKKS scheme, whose security is guaranteed by the hardness of the RLWE problem[1], ensuring polynomial-time indistinguishability. During inference, we employ only CKKS-supported secure homomorphic operations (PMult,CMult,Rot), and the ciphertext packings. Since CKKS inherently supports arbitrary message vector ordering in its plaintext polynomial encoding[2] and homomorphic operations, FicGCN preserves CKKS's original security guarantees.
> In the setup stage, we select homomorphic encryption parameters[2] to achieve 128-bit security—meaning any successful attack would require at least $2^{128}$ basic operations.
>
> **2. Analysis of the impact of supplementary references on FicGCN**
>
> The techniques from the referenced works are orthogonal to our research and do not directly contribute to it. However, they can be easily integrated with our approach in their intended applications to enhance overall end-to-end performance.
>
> To illustrate, FedML-HE enables clients to strategically compromise certain node features' security via Selective Encryption, converting partial computations from ciphertext to plaintext for reduced latency. THE-X primarily optimizes nonlinear layers through polynomial approximation—an objective orthogonal to FicGCN's linear-layer optimizations. Similarly, BatchCrypt's quantization uniformly accelerates all layers, constituting another independent optimization dimension. While all 3 studies offer potential improvements for GCN inference, their techniques operate on distinct axes from our approach. We therefore incorporate them as valuable references while emphasizing FicGCN's unique contributions.
>
>
> **3. Key management in FicGCN**
>
> In FicGCN, we exclusively employ FHE to execute encrypted inference for ensuring data security. This framework requires three distinct keys: the public key (pk), private key (sk), and evaluation key (evk).
>
> - Pk: Used for data encryption/decryption and enabling homomorphic operations, thus being publicly shared between both parties.
> - Sk:  Also involved in encryption/decryption, is strictly client-exclusive to prevent server-side privacy breaches. All decryption occurs solely on the client side to eliminate man-in-the-middle risks.
> - Evk: Generated by the client and disclosed to the server, facilitates server-side HE computations (e.g. Rotation, CMult).
>
> **4. Analysis of collusion threat model in FicGCN**
>
> FicGCN's application scenario involves strictly one client and one server following the common FHE assumption[3], where the client possesses all node features and refuses to disclose any data to the server. This fundamentally contradicts the prerequisite for collusion models defined in [4]—which require at least three participating parties capable of covert coordination to violate protocol security. Under our key management framework, **FHE inherently eliminates collusion threats** because:
>
> (1) Only two non-cooperating parties exist
>
> (2) Even if a server supports multiple colluding clients, they cannot obtain honest clients' private keys
>
> (3) Security reduces solely to RLWE hardness
>
>
>
> Reference:
>
> [1] Oded Regev. 2005. On lattices, learning with errors, random linear codes, and cryptography. In Proceedings of the thirty-seventh annual ACM symposium on Theory of computing (STOC '05)
>
> [2] Cheon, J.H., Kim, A., Kim, M., Song, Y. (2017). Homomorphic Encryption for Arithmetic of Approximate Numbers. In: Takagi, T., Peyrin, T. (eds) Advances in Cryptology – ASIACRYPT 2017.
>
> [3] Ran Ran, Nuo Xu, Wei Wang, Gang Quan, Jieming Yin, and Wujie Wen. 2022. CryptoGCN: fast and scalable homomorphically encrypted graph convolutional network inference. In Proceedings of the 36th International Conference on Neural Information Processing Systems (NIPS '22).
>
> [4] Goldreich, O. (2004). Foundations of Cryptography II. Cambridge University Press.

---

### Official Review · Reviewer_DHPE · 2025-03-14

**Overall Recommendation:** 3

**Summary:**

The paper proposes FicGCN, a framework designed to accelerate private Graph Convolutional Network inference, with three key innovations. First, an optimal layer-wise aggregation scheduling strategy is presented to accelerate inference for data of various scales. Second, Sparse Intra-Ciphertext Aggregation (SpIntra-CA) is introduced to leverage GCN sparsity to minimize the overhead associated with rotations in ciphertexts during aggregation operations. Third,  Node Order Optimization (NOO) is proposed to minimize conflicts and improve computation efficiency by reordering nodes based on the adjacency structure. FicGCN is evaluated on several popular datasets, and the results show that FicGCN achieved the best performance across all tested datasets, with up to a 4.10× improvement over the latest design.

## update after rebuttal
Based on the authors’ rebuttal, I have revised my original score from 2 to 3 since the rebuttal resolved my major concerns on the motivation of the NOO technique.

**Claims And Evidence:**

Yes

**Essential References Not Discussed:**

No, essential references are all discussed.

**Experimental Designs Or Analyses:**

Yes

**Methods And Evaluation Criteria:**

No. I have one question on the method proposed in this paper. It seems this paper mainly focuses on optimizing the node’s order so that the adjacent row in the feature map would represent neighbor nodes in the graph, which could then benefit the aggregation process. However, since the graph topology (the adjacency matrix A) is a public input to both parties, they can first negotiate and determine the optimal packing method before private inference. Moreover, Ciphertext-Plaintext computation in the combination process (XW part) would not change the node order. So I think the need to reorder the node’s packing during runtime requires further justification. (Q1)

**Other Comments Or Suggestions:**

- On section 3.3, paragraph 2, line 2, I suppose the author wants to express “and thus is inefficient” instead of “and thus is efficient”.

**Other Strengths And Weaknesses:**

Strengths:
+ Only aggregating a subset of the neighbor node is an effective way to exploit the sparsity.

Weaknesses:
- More details are required to better clarify this paper, especially questions proposed in the comments.

**Questions For Authors:**

Besides the questions (Q1 and Q2) described above, I have the following questions to the author:

- Q3: What are the crypto primitives used in the framework? Is it HE/MPC or FHE?
- Q4: In the proposed method, frequent masking is required and this will result in increased multiplication depth. How does this influence the computation efficiency of HE? Could you provide a theoretic estimation of the multiplication depth required?

**Relation To Broader Scientific Literature:**

This paper mainly focuses on private GCN inference, which is meaningful for Graph relevent tasks including recommendation system, knowledge graphs and can be further extended.

**Theoretical Claims:**

Yes. I have one question on the correctness of CPOO’s theoretical overhead (Page 5). The original paper claims the worst case requires (n-1)logN^2 rotations, but once a conflict happens, the number of rotations will double as this will split the original ciphertext into two ciphertexts, and result in like 2^logN = N ciphertext instead of logN^2.Therefore, detailed proof for this worst-case scenario is needed. Moreover, I do not find the meaning of n, and I suggest the author make a table to better illustrate the meaning of the notations. (Q2)

---

> ### Author Rebuttal · Authors · 2025-04-01
>
> **1. Justification for the need to reorder nodes(Q1)**
>
> Thanks for your comments. Negotiating and determining the optimal packing method based on NOO prior to private inference indeed constitutes a key contribution of our work.  We discussed it in Section 3.2.1 and Figure 8(b).
> However, only packing is insufficient due to the leak of utilization of sparsity of A in online inference(Table 4,6), we thus introduce three techniques and put them together to support efficient FHE-GCN inference:
> - Latency-aware packing: Optimizes packing for XW to enable dense SIMD computations.
> - SpIntra-CA: A novel approach leverages irregular sparsity patterns during AX, enabling arbitrary desire node order and supporting parallel and non-redundant node aggregation.
> - NOO: A reordering algorithm that groups nodes to enhance SpIntra-CA's computational efficiency while preserving XW performance.
>
> To illustrate, in Figure 8(b) both parties iteratively determine the node packing order, starting from the final layer and back-propagating to the client's input based on NOO and SpIntra-CA. This reverse derivation is then integrated with our latency-aware strategy to derive the optimal packing method, achieving simultaneous optimization of both AX and XW computation.
>
> **2. Proof of upper bound rotation counts in SpIntra-CA (Q2)**
>
> Due to space constraints, we present the key proof arguments and the complete formal proof with notation tables is at https://anonymous.4open.science/r/FicGCN-D208. In our paper, n denotes the sampled neighbor counts.
>
> The rotation counts of each stage in SpIntra-CA can be calculated as $\sum\_{k=0}^{i-1}Con_{k,j+\sum\_{p=0}^{k-1}2^p} \cdot \frac{1}{2^k}\leq \sum\_{k=0}^{i-1}2^k \cdot \frac{1}{2^k}$, is less than $log(N)$. Thus the total counts is $O(log^2(N))$  rather than $O(N)$. This can be formally proved by the Lemma 1 and Conflict Bound Analysis.
>
> **Analysis:** 1) $Con_{i,j}$, the number of conflicts in the $i^{th}$ rotation stage at the $j^{th}$ slot, is the summation of all conflicts during the prior stages in the corresponding slots which can be rotated to $j^{th}$.
> 2) $PR(ct[m]\rightarrow ct[q])$, the the probability of rotating an element in $m^{th}$ slot to $q^{th}$. The case "$ct[j+\sum^{k-1}\_{p=0}2^p]\rightarrow ct[j]$" means that the rotation step($rs$) bits of an element in $ct[j+\sum^{k-1}\_{p=0}2^p]$ should be all ``1'' among the corresponding k contiguous bits.
>
>
> **Lemma 1:** Given any bit distribution($Pr[bit_p=1]=\frac{1}{2}-\epsilon$),  $ PR(ct[j+\sum\_{p=0}^{k-1}2^p]\rightarrow ct[j])\le \frac{1}{2^k}$
>
> **Proof:**
> Based on **Analysis** 2), we have $PR(ct[j+\sum\_{p=0}^{k-1}2^p]\rightarrow ct[j])\le \frac{1}{2^k}$ when $\epsilon>0$. When $\epsilon\le0$, by replacing cyclic left shifts with cyclic right shifts, the $rs$ bits become the complement of the original:
> $$PR(ct[j+\sum^{k-1}\_{p=0}2^p]\rightarrow ct[j])=(\frac{1}{2}+\epsilon)^k \leq \frac{1}{2^k}$$ So we have $ PR(ct[j+\sum\_{p=0}^{k-1}2^p]\rightarrow ct[j])< \frac{1}{2^k}$ when $\epsilon\neq 0$
>
> **Conflict Upper Bound Analysis:** $Con_{i,j} \leq log(N)$ under the worst case(uniform distribution).
>
> **Proof:**  According to **Analysis** 1), $$Con_{i,j}=\sum\_{k=0}^{i-1}Con_{k,j+\sum\_{p=0}^{k-1}2^p} \cdot PR(ct[j+\sum\_{p=0}^{k-1}2^p]\rightarrow ct[j]) $$
> According to Lemma 1, it is $$\sum\_{k=0}^{i-1}Con_{k,j+\sum\_{p=0}^{k-1}2^p} \cdot \frac{1}{2^k}\leq \sum\_{k=0}^{i-1}2^k \cdot \frac{1}{2^k}=i \leq log(N)$$
> Thus, the number of conflicts per stage is bounded by log(N). Given that SpIntra-CA comprises log(N) stages, it yields a total conflict counts of less than $log^2(N)$ between an aggregation of two nodes.
>
> **3. Crypto primitives in FicGCN (Q3)**
>
> We use only FHE in FicGCN framework, with no other crypto primitives such as MPC.
>
> **4. Impact of masking on multiplicative depth**
>
> In  CKKS, the consumption of multiplicative depth primarily stems from ciphertext rescaling operations. However, the masking procedures described in our work do not require such rescaling and introduce no additional burden on the multiplicative depth budget.
>
> During CKKS, each floating-point message $m$ is first scaled by a precision parameter $\Delta$ and quantized to an integer before encryption:
> $c=Enc(\Delta \cdot m)\ $.
> Consequently, when multiplying two ciphertexts with identical initial scales ($\Delta$), the resulting ciphertext's scale becomes $\Delta^2$:
> $$c_1*c_2=Enc(\Delta\cdot m_1)*Enc(\Delta\cdot m_2)=Enc(\Delta^2\cdot(m_1\*m_2)) $$
>
> CKKS employs rescaling operations to restore the ciphertext's scale to $\Delta$. However during Masking, we multiply the ciphertexts by plaintexts encoding integer 0/1 with the scale=1:
> $$ Enc(\Delta \cdot m) * Encode(1 \cdot mask) = Enc(\Delta \cdot (m*mask))$$
> Therefore the scale of ciphertexts remains invariant. Also, the results of Masking  bit-length remains constant because we multiply them by 0/1. Thus the multiplicative depth will not be consumed during Masking.

---

### Official Review · Reviewer_B6Z3 · 2025-03-16

**Overall Recommendation:** 4

**Summary:**

The paper presents FicGCN, a method aimed at enhancing the efficiency of homomorphic encryption in irregular Graph Convolutional Networks (GCNs). It introduces a **latency-aware packing method** that optimizes the arrangement of ciphertext slots, balancing computational overhead and utilization. The **Sparse Intra-Ciphertext Aggregation (SpIntra-CA)** method is proposed to minimize redundant computations during aggregation, leveraging the sparsity of graphs for parallel processing of neighboring nodes. Additionally, the **Node Order Optimization (NOO)** technique is introduced, which rearranges nodes to reduce rotation overhead and conflicts during ciphertext processing. The paper also includes a **region-based data reordering method** that improves aggregation efficiency by organizing data based on local adjacency structures. Experimental results demonstrate that FicGCN achieves up to a **4.1x speedup** compared to state-of-the-art methods, particularly benefiting larger datasets. Overall, the study emphasizes the importance of optimizing homomorphic encryption operations to enhance computational efficiency while maintaining data privacy in GCN applications.

**Claims And Evidence:**

Yes, all claims are clear and with enough proof.

**Essential References Not Discussed:**

None

**Experimental Designs Or Analyses:**

I have checked the experiments, all looks good and enough to support the techniques.

**Methods And Evaluation Criteria:**

The Node Order Optimization (NOO) technique improves computational efficiency during encrypted data processing by determining the optimal aggregation order for each node. This minimizes the shift range within the ciphertext, which is crucial for reducing rotation overhead during aggregation operations. By arranging nodes in an aggregation-friendly manner, NOO enhances the efficiency of the Sparse Intra-Ciphertext Aggregation (SpIntra-CA) method, leading to a significant reduction in the number of required rotations and overall computational complexity. This optimization allows for more effective utilization of ciphertext slots, ultimately improving the speed and efficiency of the graph convolutional network operations. With the provided benchmarks, i think this work handles the sparsity of graph in HE domain very well.

**Other Comments Or Suggestions:**

none

**Other Strengths And Weaknesses:**

I have only one concern about the scalability to very large model. As graph model might be applied to dataset with over 100k nodes, e.g. reddit, applying node detection here might has very large cost on graph traverse.

**Questions For Authors:**

none

**Relation To Broader Scientific Literature:**

see above.

**Theoretical Claims:**

I have checked correctness of proof. No issues found by me.

---

> ### Author Rebuttal · Authors · 2025-04-01
>
> **Time complexity of Node Order Optimization (NOO) on large-scale datasets**
>
> Thanks for your constructive comments. We have extended the application of NOO to large-scale datasets  Pokec (1.6M nodes). The following table summarizes the experimental results across datasets, including:
> - **Graph Statistics**: Node count ($|V(G)|$), edge count ($|E(G)|$), and average degree ($d_{avg}$).
> - Time Overhead: NOO preprocessing time ($T_{NOO}$) and online stage FHE computation latency ($T_{FHE}$).
> - Efficiency Ratio: $\rho=T_{NOO}/T_{FHE}$
>
>
> | Dataset      | $\|V(G)\|$ | $\|E(G)\|$     | $d_{avg}$ |  $T_{NOO}$ (s)     | $T_{FHE}$ (s) | $\rho$      |
> | :-----------: | :-----------: | :-----------: | :-----------: | :-----------: | :-----------: | :-----------: |
> | Cora      | 2.7K       | 5.4K      | 4.01       | 2.00      | 64.12    | 3.2%       |
> |Citeseer     | 3.3K       |  4.7K      | 2.85       | 5.33       | 79.98      | 6.7%       |
> | Corafull      | 19.8K       |  127K     | 12.82       | 27.91      | 7733      | 0.36%       |
> |Pokec      | 1.63M       |  30.62M      | 18.80      | 181.20       | $\sim 10^7$     | $\sim 0.002$%      |
>
>
> As shown in this table, NOO exhibits low time overhead on smaller datasets. For large-scale graphs like Pokec,  NOO's overhead remains negligible relative to FicGCN's online phase. Notably, on large graphs, the client-side encryption time alone eclipses NOO's latency, ensuring NOO does not bottleneck FicGCN's performance.

---

### Decision · Program_Chairs · 2025-05-01

**Decision:**

Accept (poster)

**Comment:**

This paper proposes a homomorphic encryption based framework for GNNs that exploits sparsity. This includes in particular a scheme for sparsity-aware aggregation, a latency-aware packing method that optimizes combination operations, and a node-order optimization. The empirical evaluation shows an up to 4x speedup over existing methods.
The reviewers appreciated the contributions, especially the efficiency gains due to exploiting the sparsity of graphs. One reviewer asked for a security proof, please include the argumentation in the paper, as well as relation to the additional related works.